# Appetitive Motivation and Associated Neurobiology Change Differentially across the Life Course of Mouse Offspring Exposed to Peri- and Postnatal High Fat Feeding

**DOI:** 10.3390/nu14235161

**Published:** 2022-12-04

**Authors:** Laura Contu, Christopher J. Heath, Cheryl A. Hawkes

**Affiliations:** 1School of Life, Health and Chemical Sciences, The Open University, Milton Keynes MK7 6AA, UK; 2Division of Biomedical and Life Sciences, Lancaster University, Lancaster LA1 4YW, UK

**Keywords:** maternal obesity, ageing, food reward, neurobiology, offspring, mouse, dopamine

## Abstract

Alterations in neural pathways that regulate appetitive motivation may contribute to increased obesity risk in offspring born to mothers fed a high fat (HF) diet. However, current findings on the impact of maternal obesity on motivation in offspring are inconclusive, and there is no information about the long-lasting effects in aged animals. This study examined the longitudinal effect of perinatal and chronic postnatal HF intake on appetitive motivation in young and aged offspring. Female C57Bl/6 were fed either a control (C) or HF diet before mating through to lactation. At weaning, offspring were maintained on the C or HF diet, generating the following four diet groups: C/C, C/HF, HF/C, and HF/HF based on the pre/post weaning diet. At 6 months, motivation was higher in HF/C females, but lower in male and female C/HF and HF/HF mice. By 12 months, this difference was lost, as C-fed animals became less motivated, while motivation increased in HF-fed mice. The mRNA levels of dopamine receptor 1 and 2 increased with age, while cannabinoid receptor 1 and μ-opioid receptor expression remained stable or decreased in mesolimbic and mesocortical dopaminergic pathways. Results from this study suggest that perinatal and chronic postnatal HF feeding produced opposite effects on appetitive motivation in young adult offspring mice, which was also reflected in the shift in motivation over time. These results have significant implications for patterns of hedonic eating across the life course and the relative risk of obesity at different time points.

## 1. Introduction

The latest estimates of global obesity suggest that more than 37% of men and 30% of women are overweight or obese [1]. Rates of maternal obesity have also risen significantly over the past 10 years [2,3,4,5] due to both greater numbers of overweight pregnant women and excess weight gain during pregnancy [6,7]. It is now established that maternal obesity can adversely affect the long-term health of the offspring, including increased risk of diabetes, coronary heart disease, asthma, stroke, lower IQ, and premature death [8,9,10]. In addition, obesity during gestation is linked to increased body mass index (BMI) of offspring at birth and across childhood, adolescence, and adulthood [11,12,13,14].

Total fats account for 30–45% of total dietary intake among adults across the USA and Europe [15,16]. Current hypotheses propose that overconsumption of palatable foods results either from a blunted reward system or increased salience for the rewarding foods that override satiety signals [17,18]. Consequently, numerous animal studies have examined the effect of high fat (HF) foods on motivation for feeding and appetitive reward. While some studies have reported increased motivation in adolescent and young adult rodents fed a HF diet [19,20], others have found the opposite effect [21,22]. Similarly, early experiments using animal models of maternal obesity reported increased motivation for a palatable reward in young adult offspring born to dams fed a HF diet [23,24]. However, this has been challenged by other studies that have reported decreased appetitive motivation in offspring of obese mothers [25,26]. In humans, children of parents who are overweight or obese have a higher preference for fatty foods and consume more food than those of normal weight parents [27,28]. Despite evidence that dietary habits established during childhood and adolescence persist into adulthood [29], few animal studies have examined the effect of chronic HF feeding (e.g., >12 months) on motivation in older animals. Moreover, very little is known about the long-term effects of perinatal HF exposure on motivation across the lifespan.

Dopamine (DA) is a key regulator of motivation for hedonistic rewards and signals. This is mediated primarily by dopaminergic signaling within the mesolimbic and neocortical pathways. The mesolimbic pathway sends projections from the DA cell bodies in the ventral tegmental area (VTA) and innervates the nucleus accumbens (NAc), caudate–putamen (CPu), and limbic lobe, while the mesocortical pathway innervates the prefrontal cortex (PFC). Modulation of DA concentrations and activation of dopamine receptors 1 and 2 (D1R and D2R) within these pathways mediate different aspects of the “liking” and “wanting” of palatable foods. For example, anticipation and consumption of a food reward is associated with increased DA release in the PFC, but not in the NAc [30,31]. By contrast, modulation of DA levels in the NAc affects food-reinforced responding but does not alter total food intake [30,31]. Within the CPu, overexpression of DR2 is linked with reduced motivation for an appetitive reward in mice [32]. Thus, changes in DA concentrations, dopamine transporter (DAT) and DR1 and DR2 expression in the PFC, NAc, and CPu are thought to underlie behaviour associated with the seeking and consumption of palatable foods.

In addition to DA, motivation for appetitive reward is also influenced by the endocannabinoid (eCB) and opioid systems. The eCBs contribute to the regulation of DA signaling within the mesocortical pathway via a negative feedback loop. In this loop, dopaminergic activation of GABAergic interneurons in the PFC suppresses the activity of glutamatergic neurons that innervate the VTA. The GABA receptor activation stimulates the release of eCBs, which retroactively diffuse across the synapse and bind to cannabinoid receptor 1 (CB1) on presynaptic GABA neurons, thereby blocking GABA release [33]. The end result of CB1 stimulation is increased DA concentrations in the PFC, while antagonism of CB1 attenuates DA release induced by palatable foods [34,35]. The eCBs also modulate DA release in the NAc [36].

Separate to the modulation of DA, opioids also directly regulate food intake. Administration of opioid agonists into the NAc stimulates the ingestion of palatable foods, and μ-opioid receptor activation contributes to both the hedonistic and incentive value of rewarding stimuli [37]. Interestingly, there is some evidence that μ-opioid receptor activation contributes to appetitive motivation independently of homeostatic feeding, suggesting a specific role in the intake of pleasurable foods [38] Collectively, coordination between DA, opioid, and eCB signals within the mesolimbic and mesocortical pathways regulates appetitive motivation.

Previous studies have reported inconsistent effects of both pre- and postnatal HF feeding on DA, opioid, and eCB markers. For example, levels of D1R and D2R, CB1, and μ-opioid receptor have been found to be unchanged, upregulated, and decreased in the VTA, NAc, and/or PFC of HF-fed rodents and offspring born to mothers fed a HF diet [20,22,24,39,40,41]. Moreover, changes in these markers differ between male and female offspring [39], suggesting that there is a sexual dimorphism in the effect of perinatal obesity on reward pathways. In addition, Ong et al. reported an age-related shift in the expression of μ-opioid receptors and DA transporters between offspring born to mothers fed a cafeteria diet and those fed a control diet [42]. Thus, a definitive and persistent effect of HF feeding on key neurobiological pathways underlying appetitive motivation has not been established.

The purpose of this study was to characterize the effect of perinatal and postnatal HF diet exposure on appetitive motivation in male and female mice longitudinally across their lifespan and to characterize any related effects on the expression of DA, opioid, and eCB markers in associated brain areas.

## 2. Materials and Methods

Animal model—C57Bl/6 mice were obtained from the Biomedical Research Unit at the Open University and kept on a 12 h light/dark cycle (lights on at 7 am). Female C57Bl/6 dams were fed either a control (C, 70% kcal carbohydrate, 20% kcal protein, 10% kcal fat, *n* = 34) or high fat (HF, 45% kcal fat, 35% kcal carbohydrate, 20% kcal protein); Special Diet Services, UK, *n* = 39) diet for 4 weeks before mating (when dams were 18–27 weeks old). Dams were kept on the diet during gestation and lactation. Macro minerals, vitamins, and amino acid composition was matched between diets, as previously described [43]. Studs were maintained on the C diet until breeding. Successful delivery rate was 88% for C-fed dams and 80% for HF-fed dams. Two litters from HF-fed dams were cannibalized, but all other pups survived.

At weaning (P28), male and female offspring were randomly assigned either the C or HF diet, generating the following four experimental groups: C/C, C/HF, HF/C, HF/HF, representing the pre- and post-weaning diet, respectively. Offspring were group-housed (average *n* = 4 mice/cage) by litter and kept on the post-weaning diet until sacrifice. Mice were allowed ad libitum access until the start of the behavioral experiments. All experiments were approved by the Open University Animal Welfare and Ethics Review Board and the UK Home Office (PPL 70/8507).

Testing protocol—Motivation was tested in Bussey–Saksida Mouse Touch Screen Chambers (Campden Instruments, Loughborough, UK). Strawberry milkshake (Yazoo^®^, FrieslandCampina UK, Horsham, UK) was used as the appetitive operant reinforcer. One week before the start of the testing period, mice were food restricted to ~90–85% of their free-feeding weight and were kept on this restriction for the duration of testing. After testing was complete, animals were again allowed ad libitum access to their diet. Details of the schedule of restriction and ad lib feeding have been published previously [43].

Offspring were divided into two groups, as shown in Figure 1. All animals began testing at 6 months of age and underwent 2 days of habituation in the chambers (20 min/day with free consumption of milkshake from the reward magazine) and then an initial touch protocol (15 trials over 60 min) [44]. Following this, offspring were divided into two groups, as follows: animals in group A did not undergo additional behavioural testing and were sacrificed at 10 months of age for tissue collection (Figure 1); mice in group B underwent a longitudinal assessment protocol in which they were tested sequentially on the fixed-ratio (FR), progressive ratio (PR), and paired visual discrimination (PVD) tasks. Animals were rested until 12 months of age and then run through FR5 before being re-tested on the PR2 schedule (Figure 1). Tissues were collected from group B mice when the animals were 16-months-old. The number of mice tested in each group is shown in Figure 1. Data obtained from the PVD task are presented in a separate manuscript (in preparation).

FR and PR tasks—Touchscreen FR and PR schedules were carried out as described previously [44], with some modifications. Testing was carried out during the light phase, between 10:00 and 15:00. In brief, animals were trained progressively through FR1 (15 trials), FR2 (15 trials), FR3 (10 trials), and FR5 (6 trials) sessions. Mice that did not reach the criteria in FR5 (6 trials completed in 60 min) were excluded from further testing. After completion of FR5, animals were evaluated in a PR2 schedule, in which the reward response requirement was increased on a linear + 2 basis upon completion of each trial (e.g., PR2 = 1, 3, 5, 7, etc. touches per reward) over a 60 min period. The session was terminated if no response was detected within 5 min, and breakpoint was determined by the number of target responses emitted in the last completed trial. Breakpoints were averaged over 3 sequential days. Data corresponding to front and rear beam breaks/sec, magazine entries/sec, and latency to collect reward were also collected and averaged over the 3 days.

Sucrose preference—After completion of the behavioral tasks, mice were allowed ad lib access to their assigned diet for 5 days. Subsequently, mice were singly housed for 24 h and then randomly assigned to receive milkshake or 9.6% sucrose (matched to the sucrose concentration in strawberry Yazoo) alongside assigned diet and water in their home cages for 24 h, and intake was recorded. The next day, mice were given the opposite liquid so that all mice were exposed to both sucrose and milkshake. Placement of water and sucrose/milkshake was counterbalanced to reduce any potential side bias. Animals were then re-housed with their littermates until sacrifice.

RT-qPCR—Tissues were collected during the light phase. Mice were perfused with 0.01 M PBS (*n* = 5/group/age) after being deeply anesthetized with an overdose of sodium pentobarbital. Brains were rapidly dissected, and the PFC, CPu, and NAc were snap frozen on dry ice and kept at −80 °C until use. Frozen tissues were placed in RNAlater-ICE (Fisher Scientific, Loughborough, UK) and transferred to −20 °C for at least 2 days before being processed for RNA extraction using the RNeasy Plus Universal Mini Kit (Qiagen, Manchester, UK) according to the manufacturer’s instructions. The cDNA was synthesised from 10 ng RNA using the Applied Biosystems™ High-Capacity cDNA Reverse Transcription Kit (Fisher Scientific, Loughborough, UK). KiCqStart^®^ SYBR^®^ Green Primers (Merck, Gillingham, UK) were used to quantify levels of cannabinoid receptor 1 (*Cnr1*), μ-opioid receptor 1 (*Oprm1*), dopamine receptor 1 (*Drd1a*), dopamine receptor 2 (*Drd2*), and β-actin (*Actb*), using the QuantiTect SYBR Green PCR Kit (Qiagen). Primer sequences and cycling parameters are shown in Appendix A. The mRNA expression of the gene of interest was normalised to β-actin. Relative gene expression was quantified using the 2^−ΔΔCt^ method. For comparison between diet groups, C/C was used as the reference group, while male animals were used as the reference group for comparison between sexes. For age-related comparisons, values from the 16-month-old mice were used as the reference group.

Western blotting—Frozen brain tissues (*n* = 5/group/age) were homogenized in RIPA lysis buffer [150 mM NaCl, 0.1% SDS, 50 mM NaF, 1 mM NaVO3, 20 mM Tris-HCl (pH 8.0), 1 mM EDTA, 1% Igepal] containing a protease inhibitor cocktail (Merck Millipore, Watford, UK). Homogenates were centrifuged (10,000× *g*, 10 min, 4 °C), and supernatants were frozen at −80 °C until further use. Proteins (15 μg) were separated by gel electrophoresis using 4–20% Tris-glycine gels (Fisher Scientific, Loughborough, UK), and then transferred onto a nitrocellulose membrane. Membranes were incubated overnight at 4 °C with anti-dopamine transporter antibody (DAT, 1:750, Merck, Feltham, UK). Protein loading was confirmed by stripping and re-probing membranes with anti-glyceraldehyde-3-phosphate dehydrogenase (GAPDH, 1:50,000, Merck). Data was generated from replicates of two blots. Densitometry of immunoblots was quantified using Image J (NIH, Bethesda, MD, USA) and calculated as an optical density ratio of protein levels normalized to GAPDH levels.

Statistical analysis—Data with *n* > 20 samples/group were checked for normality using the Kolmogorov–Smirnov test, while data with *n* < 20 samples/group were checked using the Shapiro–Wilk test. Outliers were identified and excluded using the ROUT test. Analyses between diet groups were carried out using one-way ANOVA with Sidak’s post hoc test, or a Kruskal–Wallis test with Dunn’s post hoc test where data were non-parametric. Evaluations of sex x diet were carried out using two-way ANOVA with Sidak’s post hoc test. For age x diet analyses, only animals for which there was information at both time points were included, and comparisons were carried out using two-way, repeated measures ANOVA with Sidak’s post hoc test. All analyses were carried out using GraphPad Prism (San Diego, CA, USA). Data represent mean ± SEM unless otherwise specified, and *p* < 0.05 was considered to be statistically significant.

## 3. Results

### 3.1. High Fat Diet Induced an Obese Phenotype in Dams and Offspring

After 4 weeks of feeding, HF-fed dams weighed significantly more than C-fed dams, and this was maintained during gestation and after weaning (Figure 2A). No differences were observed in litter size, or in the male-to-female ratio of offspring born to C- and HF-fed mothers (Figure 2B,C). Male and female C/HF and HF/HF offspring weighed significantly more than C/C and HF/C animals within 3 weeks post-weaning (Figure 2D,E) and this weight difference was maintained throughout the experiment until sacrifice at 10- and 16-months-of-age (Figure 2F,G). A significant age-related weight gain was observed in C/HF males and HF/HF females, but body weight was constant between 10- and 16-months in the other offspring groups (Figure 2H,I). The HF-fed animals also had significantly more peri-gonadal fat than C-fed offspring at both ages (Figure 2J,K); however, only female C/HF and HF/HF mice showed an age-related increase in peri-gonadal fat weight (Figure 2L,M). Additional information about dam and offspring body weight, as well as food and calorie consumption, has been published previously [43].

### 3.2. Young HF-Fed Offspring Were Less Motivated for Milkshake Than C-Fed Offspring

To determine the long-term effects of peri- and postnatal obesity on motivation for an appetitive reward, male and female offspring were tested in the PR2 task at 6 months of age and again when they were 12 months old. In young animals, breakpoint was significantly lower in both male and female C/HF and HF/HF mice compared to C/C and HF/C animals, respectively (Figure 3A,B). Latency to collect reward was also significantly higher in the C/HF and HF/HF groups (Appendix A). Maternal diet had no effect on breakpoint in male offspring (Figure 3A); however, female HF/C mice had a significantly higher breakpoint than female C/C mice (Figure 3B). Breakpoint did not differ between male and female offspring in any diet group (Figure 3C). Magazine entries/sec was similar between male diet groups; however, entries were decreased in C/HF vs. C/C female mice (Appendix A). In addition, female C/C and HF/HF mice made significantly more magazine entries/sec than males in the same diet groups (Appendix A). The number of front beam breaks/sec was significantly decreased in C/HF male mice compared to C/C males, but no differences were observed between female diet groups (Appendix A). In addition, female mice made significantly fewer front beam breaks/sec than male mice in the same diet groups except for C/HF females, where the difference bordered significance (*p* = 0.05; Appendix A). No differences in rear beam breaks/sec were observed between male or female diet groups, except for fewer breaks in HF/HF males vs. HF/C males (Appendix A). Rear beam breaks/sec did not differ between any male vs. female groups (Appendix A).

### 3.3. Motivation for Milkshake Reward Did Not Differ between Aged C- and HF-Fed Offspring

At 12 months of age, breakpoints were similar between all diet groups in both male and female offspring (Figure 3D–F). Reward collection latencies were also unaltered between diet groups in both sexes (Appendix A). Magazine entries/sec of males were similar between diet groups (Appendix A). Female C/HF offspring made significantly fewer magazine entries/second than C/C females, while entries/sec were similar between males and females in all diet groups (Appendix A). Front beam breaks/sec were significantly lower in HF/HF vs. HF/C males, and in C/HF vs. C/C females; however, only female C/HF and HF/C offspring showed significantly fewer breaks compared to males in the same diet group (Appendix A). The number of rear beam breaks/sec did not differ between any diet groups or between males and females (Appendix A).

### 3.4. Motivation Decreased over Time in C-Fed Offspring but Increased in HF-Fed Animals

Performance over time showed a decrease in breakpoint between 6 and 12 months of age in male C/C and HF/C mice, although this was only significant in the HF/C group (Figure 3G). By contrast, breakpoint was significantly increased in 12-month-old C/HF males compared to their performance at 6 months of age. The HF/HF animals also showed a non-significant increase in breakpoint with age. Reward collection latency was not altered between young and aged males (Appendix A), although magazine entries/sec were significantly increased in 12-month-old C/C males (Appendix A). No differences were observed in front or rear beam breaks/sec between 6- and 12-month-old males in any diet group (Appendix A).

In females, breakpoint remained stable between 6- and 12-month-old C/C animals, while changes in the other diet groups had a similar but non-significant pattern to that of male offspring (Figure 3H). Latency to collect reward was faster in aged C/HF females (Appendix A). Magazine entries/sec did not differ between 6- and 12-month-old female mice in any diet group (Appendix A). Female C/C mice showed a significant age-related increase in front beam breaks/sec (Appendix A), while rear beam breaks/sec were unaltered between 6 and 12 months of age in any diet group (Appendix A).

### 3.5. HF-Fed Mice Showed Preference for Milkshake over Sucrose

To determine if differences in appetitive motivation were related to differential taste preferences, fully fed 9- and 15-month-old mice were given ad lib access to water, milkshake, and sucrose for 24 h after completion of the behavioural tasks. All animals drank significantly less water than sucrose or milkshake, regardless of age or diet group (Figure 4A–F). Both male and female C/C and HF/C mice drank similar amounts of sucrose and milkshake at 9-months of age. However, all HF-fed animals consumed significantly more milkshake than sucrose (Figure 4A,B). Comparison of intake between diet groups found that milkshake consumption was similar between all groups; however, C/HF and HF/HF mice drank less sucrose compared to C/C and HF/C animals, respectively (Figure 4A,B). No sex differences were observed in the consumption of sucrose and milkshake in any diet group (Figure 4C).

At 15 months of age, male mice drank the same amount of milkshake and sucrose regardless of pre- and postnatal diet (Figure 4D). In females, both C/HF and HF/C mice drank significantly more milkshake than sucrose, and a similar non-significant pattern was also observed in the C/C and HF/HF females (Figure 4E). Analysis of diet-related differences showed that C/HF and HF/HF females drank significantly less sucrose and milkshake than C/C and HF/C females, respectively. Aged males and females consumed similar amounts of both sucrose and milkshake regardless of diet group (Figure 4F). Compared to at 9 months of age, 15-month-old male and female C/C animals consumed significantly more milkshake; however, no age-related differences in sucrose or milkshake were noted in the other diet groups (Figure 4G–J).

A summary of offspring body weight, motivation, and sucrose preference in young and aged offspring is shown in Table 1.

### 3.6. Expression of Markers of DA, μ- Opioid and CB1 Receptors between Diet Groups in 10-Month-Old Offspring

To evaluate the impact of pre- and postnatal HF diet on expression of genes in neural circuits associated with reward and motivation, mRNA levels of *Cnr1*, *Drd1a*, *Drd2*, and *Oprm1* were evaluated in the NAc, CPu and PFC of 10- and 16-month-old offspring. Because DAT mRNA expression is largely limited to the ventral tegmental area (45), DAT protein expression was determined in the NAc, CPu and PFC by western blotting.

In 10-month-old mice, mRNA levels of the genes of interest were generally unaltered between diet groups. Within the NAc, there was a significant increase in *Cnr1* expression in HF/HF males compared to C/HF animals (Figure 5A). No differences in *Cnr1* were observed between female diet groups (Figure 5A). Levels of *Drd1a*, *Drd2*, and *Oprm1* were also unchanged between diet groups (Appendix A). The DAT protein levels were significantly decreased in C/HF and HF/C males compared to C/C males (Figure 6A), while no differences were observed between female diet groups (Figure 6A).

In the CPu, *Drd2* expression was significantly increased in HF/HF males vs. HF/C males but did not differ between other male diet groups or between female groups (Figure 5B). Diet did not influence the levels of *Drd1a*, *Cnr1*, and *Oprm1* in the CPu (Appendix A). Similarly, DAT expression did not differ between male or female diet groups (Figure 6B). There were no differences in *Drd1a*, *Drd2*, *Cnr1,* or *Oprm1* expression between diet groups in the PFC (Appendix A). The DAT levels were also unchanged between diet groups in the PFC (Figure 6C).

### 3.7. Expression of Markers of DA, μ-Opioid and CB1 Receptors between Diet Groups in 16-Month-Old Offspring

In 16-month-old animals, levels of *Drd1a* were increased in the NAc of HF/C mice vs. C/C males (Figure 5C). A similar but non-significant pattern was seen between the female diet groups. Expression of *Drd2, Crn1*, and *Oprn1* in the NAc was not altered by diet in either male or female offspring (Appendix A). No differences were observed in *Drd1a, Drd2, Oprn1*, or *Crn1* levels between diet groups in the CPu of males or female mice (Appendix A). In the PFC, expression of all genes of interest was stable between male diet groups (Figure 5E,F and Appendix A). In female mice, *Drd1a* levels were significantly elevated by 4–5-fold in C/HF mice compared to C/C females (Figure 5E). The *Oprm1* and *Cnr1* levels were unaltered between diet groups in the PFC (Appendix A). The DAT protein levels did not vary between diet groups in the NAc, CPu, or PFC of aged animals (Figure 6A–C).

### 3.8. Expression of Markers of DA, μ-Opioid and CB1 Receptors between Male and Female Offspring at Both Ages

In 10-month-old animals, no differences were observed between male and female offspring in the expression of *Crn1, Drd1a, Drd2, Oprm1* (Figure 5A and Appendix A), or DAT protein levels in the NAc (Figure 6A). In the CPu, *Drd2* levels were significantly higher in female C/C and HF/C mice compared to males in the same groups (Figure 5B). However, sex did not influence the levels of *Drd1a*, *Cnr1*, and *Oprm1* (Appendix A). Similarly, there were no differences in *Drd1a*, *Cnr1*, or *Oprm1* expression between male and female offspring in the PFC (Appendix A). The *Drd2* expression was increased in HF/HF females compared to HF/HF males; however, this bordered significance (*p* = 0.05, Appendix A). The DAT protein levels were unaffected by sex differences in both the CPu and PFC (Figure 6B,C).

In 16-month-old animals, levels of *Drd1a, Drd2*, and *Crn1* in the NAc were similar between male and female offspring (Appendix A), but levels of *Oprm1* were significantly higher in HF/C females compared to males in this diet group (Figure 5D). In the Cpu, female HF/C mice expressed higher levels of *Drd1a* than male counterparts (Appendix A). There was no difference between males and females in the expression of *Drd2, Oprn1*, or *Crn1* in the aged CPu (Appendix A). In the PFC, *Drd1a* levels were significantly elevated by 4–5-fold in C/HF females compared to C/HF males (Figure 5E). A significant 7-fold increase in *Drd2* levels was also noted in the PFC of C/HF females vs. C/HF males (Figure 5F). Furthermore, *Oprm1* and *Cnr1* levels were unaltered between males and females (Appendix A). There were also no differences in DAT protein levels between males and females in the NAc, CPu, or PFC of aged animals (Figure 5A–C).

### 3.9. Expression of DA, μ-Opioid and CB1 Receptors Is Altered between Young and Aged Animals

Evaluation of changes in gene expression in the NAc between 10 and 16 months found a significant decrease in *Drd1a* levels in aged C/C males, but no significant differences in other male diet groups or between female groups (Figure 7A). By contrast, *Drd2* levels were significantly upregulated in all aged male groups except C/C mice, including a 19-fold increase in the HF/HF mice (Figure 7B). The *Drd2* expression also significantly increased by 5- to 11-fold between 10 and 16 months of age in females in all diet groups (Figure 7B). The *Crn1* expression in the NAc was stable between ages, while *Oprm1* levels were decreased in 16-month-old HF/C and HF/HF males, although this was only significant in HF/HF animals (Appendix A).

In the CPu, levels of *Drd1a* were significantly increased in all aged female animals and in all aged male diet groups except C/HF mice (Figure 7C). The *Drd2* expression was also upregulated in 16-month-old C/C and HF/C males and in C/HF female mice (Figure 7D). An opposite pattern of expression was observed for *Oprm1* and *Crn1.* The *Oprm1* levels were significantly decreased in 16-month-old C/C, C/HF, and HF/C males, with a similar non-significant trend observed in HF/HF males (Figure 7E). The *Crn1* expression was also decreased in aged C/C and C/HF male mice (Figure 7F). In female mice, there was a non-significant trend towards decreased expression of *Oprm1* in all diet groups and a significant decrease in *Crn1* in all aged diet groups, except in C/HF animals (Figure 7E,F).

In the PFC, *Drd2* levels were decreased by 3-fold in aged C/HF males and by 4-fold in C/C females (Figure 7G). A similar pattern of change was observed in *Drd1a* mRNA; however, the differences were not statistically significant (Appendix A). While *Oprm1* levels in the PFC were stable across time (Appendix A), *Crn1* expression was decreased in 16-month-old HF/HF males and in aged C/C, HF/C, and HF/HF females (Figure 7H). The DAT levels were stable across time in the NAc and Cpu of male and female mice (Figure 7A,B). In the PFC, DAT levels were decreased in 16-month-old C/HF, HF/C, and HF/HF males; however, these changes bordered significance (*p* = 0.05, Figure 6C). A summary of gene and protein changes in relation to age and diet is shown in Table 2.

## 4. Discussion

Although maternal HF feeding is known to have lasting effects on the offspring brain, few studies have examined the impact of prenatal HF feeding on offspring behaviour beyond early adulthood. In addition, most postnatal diet-induced obesity studies use relatively short feeding periods that do not model human patterns of food consumption. This study examined the impact of perinatal exposure to HF diet alone and in combination with chronic postnatal HF intake on motivation in young and aged offspring and changes in the associated neurobiology. Our results suggest that young adult female, but not male, HF/C mice had higher levels of motivation than offspring born to C-fed dams. Male and female C/HF offspring had significantly decreased motivation, irrespective of maternal diet. However, these differences were no longer observed at 12-months of age, due to a general decrease in motivation in C-fed animals and an increase in motivation in HF-fed mice. In parallel, there was an age-related increase in *Drd1a* and *Drd2* mRNA expression, and stable or decreased expression of *Crn1* and *Oprm1* in the mesolimbic and mesocortical DA pathways.

Initial studies using rat dams fed a 20–30% fat diet before and/or during gestation and lactation found that young adult male offspring were more motivated for a food reward than offspring of dams fed a 5% fat diet [23,24]. However, experiments in which mouse pups were exposed to diets with a very high fat content during gestation (60% fat), reported a lower breakpoint in adulthood compared to offspring born to mothers fed a 22% fat diet [25,26]. These discrepancies may relate to differences in source and content of fat, onset/duration of maternal feeding, age, species, and body composition of offspring at testing. The current study suggests that gestational and early life exposure to a diet with 45% fat increases motivation for an appetitive reward in 6-month-old HF/C female mice.

By contrast, appetitive motivation was not altered in male HF/C offspring. This provides further evidence of sexual dimorphism in the sensitivity of the fetal brain to maternal diet [45,46,47,48]. In addition, we found that young female mice generally made fewer front beam breaks/sec and more magazine entries/sec than male animals. This suggests that male and females engaged differently with the PR task, which may be related in part to differential effects of sex hormones on DA production and regulation of D2R and DAT levels in the CPu, NAc, and PFC [49,50]. We found that *Drd2* mRNA levels were significantly higher in the CPu of 10-month-old HF/C females vs. HF/C males. Prolonged DA release can induce desensitization of D2R autoreceptors in the CPu and NAc, leading to burst firing of dopaminergic neurons [51,52]. More work is needed to determine if these mechanisms contribute to the sex differences in motivation observed in the current study.

Parental dietary habits are known to influence their children’s food preferences and eating patterns, which contribute to risk of obesity [53,54]. Our results suggest that adult mice maintained on the HF diet throughout the post-weaning period had significantly decreased appetitive motivation at 6 months of age. Moreover, findings that HF/HF mice had a similar breakpoint to C/HF animals and a lower breakpoint than HF/C mice suggest that prolonged postnatal HF feeding attenuated the impact of maternal diet on offspring motivation at this age. Beam breaks and consumption of milkshake were similar between diet groups, indicating that the differences in breakpoint were not due to obesity-related impairments in movement or a dislike of the milkshake reward. In addition, this was not due to difficulties in learning the task, because only animals that successfully completed FR training were progressed onto the PR task. Decreased motivation has previously been reported in mice fed a similar 45% fat diet for 4 or 12 weeks [21,22]. However, other studies have found no effect or increased motivation in rodents fed 60% or 30% fat diets, respectively [20,55]. In addition, Figlewicz et al. [19] reported that rats fed a 30% fat diet for 3 weeks during adolescence had higher motivation for sucrose, but this effect was lost when animals were subsequently switched to a normal chow diet. These results broadly mimic those observed in prenatal HF studies and suggest that exposure to a ~30% fat diet increases motivation while consumption of diets with higher fat content (e.g., ≥45% fat) decreases motivation. Whether the differential impact of fat content on motivation is due to graded adaptations in DA signaling or differences in peripheral factors, such as rate of weight gain, adiposity, and/or sensitivity to satiety hormones [56] remains to be determined.

Based on the role of DA, opioids, and eCBs in the regulation of motivation and consumption of rewarding stimuli [30,31,37], we hypothesized that markers of these systems would be altered in the mesolimbic and/or mesocortical regions of mice exposed to the HF diet. However, only levels of DAT were significantly altered between diet groups, while expression of *Drd1a*, *Drd2*, *Cnr1*, and *Oprm1* were similar in the PFC, NAc, and CPu of 10-month-old animals. As DAT regulates the reuptake of DA, the observed decrease in DAT in the NAc of C/HF and HF/HF mice may reflect a compensatory response to blunted DA transmission in response to a food reward. Previous findings have shown that cafeteria or HF feeding over a 15–16-week period attenuated DA release in the NAc and CPu in response to intake of low-calorie foods [57,58]. Moreover, in the same study, Tellez et al. reported that HF-fed mice consumed less of the low-caloric food, but the same amount of high calorie food as mice fed a low-fat diet [57]. This is similar to the pattern of consumption of sucrose and milkshake that we observed in the HF-fed mice. Given the interrelationship between DAT and D2R autoreceptors in the regulation of synaptic DA concentrations, it is not clear why levels of *Drd2* were unaltered in C/HF or HF/HF animals. However, HF feeding has been inconsistently associated with both up- and downregulation of D1R and D2R expression across the VTA, CPu, NAc, and PFC [20,40], including null findings similar to those of the current experiment [24]. Taken together, the present results suggest that long-term HF feeding may attenuate acute DA release in response to a hedonistic reward without altering DA receptor expression.

Evidence from human studies suggest that dietary patterns established in early childhood are retained into adolescence and young adulthood [29,59,60,61]. To our knowledge, this is the first rodent study to examine the longitudinal effects of perinatal diet on motivation in aged offspring animals or to use a postnatal HF feeding schedule beyond 12 months. One of the most significant observations from these novel experiments is the shift in motivation between 6- and 12-months of age in both C- and HF-fed offspring. Aging is associated with reduced food cravings in humans [62] and decreased motivation and reduced striatal D2R availability in male C57Bl/6 mice [63,64,65,66]. Thus, the shift towards decreased motivation in the 12-month-old C/C male mice is consistent with a natural age-related decline. Interestingly, motivation also decreased between 6 and 12 months of age in HF/C animals. Previous work by Ong et al. [42] found that the relative expression of *Oprm1* and *Slc6a3,* which encodes the DAT protein, in the NAc of offspring born to dams fed a control or cafeteria diet switched between 6 weeks and 3 months of age. Taken together, these data suggest that alterations in DA signaling in adolescent or young adult offspring born to HF-fed mothers may not persist into old age. Additional longitudinal studies using aged offspring are needed to confirm this hypothesis.

Contrary to the C-fed mice, motivation increased between 6- and 12-months of age in C/HF and HF/HF mice. Interestingly, the increase in breakpoint in aged HF/HF mice was approximately 50% of that in the C/HF animals. This suggests that adaptations to perinatal HF exposure more strongly counterbalance the effects of postnatal HF diet in older animals. Alterations in satiety hormones, resulting from either increased adiposity or diet-induced depletion, have been proposed to contribute to decreased appetitive motivation following prolonged HF feeding [21,57]. As the mice were kept on the same postnatal diet until sacrifice and peri-gonadal fat weight was stable between 10 and 16 months of age in C/HF and HF/HF male offspring, it is unlikely that adaptations in satiety signals contributed to the shift in motivation. Alternatively, we hypothesize that the initial exposure to the milkshake during the first testing period at 6 months of age induced a sensitization-like effect in HF-fed animals that resulted in a higher breakpoint during the second testing period. This is supported by previous findings that rats fed a HF diet had decreased motivation for an appetitive reward if first exposed to the reward during the testing period, but not when animals were repeatedly presented with the reward in their home cage before the start of testing [67]. More recently, it has been reported that mice switched to a low-fat diet after 10 weeks of HF feeding showed higher consumption of a food reward compared to the intake before the dietary switch [68]. This suggests that timing of reward presentation affects its incentive salience in HF-fed animals.

Coactivation of striatal D1R and D2R and hypersensitivity of D1R in the CPu contribute to sensitization to drugs of addiction [69,70]. Therefore, the observed age-related increase in *Drd1a* and *Drd2* gene expression in the CPu in C/HF and HF/HF animals may have contributed to the increased motivation in these animals. However, as this upregulation was observed across all diet groups, other factors must also contribute to the age-related difference in motivation between C and HF groups. Food restriction itself is known to increase the sensitivity of D2R activation and induce burst firing of DA neurons in response to cocaine [71,72]. It also attenuates age-related loss of D2R binding in chow and HF-fed animals [73,74]. Although all mice underwent a similar degree of food restriction [43], the extent to which this influenced satiety and hedonistic feeding pathways may have differed between C- and HF-fed animals. In addition, as tissues were collected approximately 6 weeks after animals were placed back on ad lib feeding, some acute FR-induced changes in gene expression may have been missed. Additionally, because mice in group A did not undergo the full battery of behavioural tests, the changes in gene expression between 10 and 16 months of age may also be influenced by differences in the degree of behavioural characterization between the two groups.

## 5. Conclusions

In conclusion, results from this study suggest that perinatal and chronic postnatal HF feeding produced opposite effects on appetitive motivation in young adult offspring mice, and this was also reflected in the shift in motivation over time. The impact of maternal HF on offspring motivation varied across the lifespan and was influenced by postnatal diet, age, sex, and possibly frequency or order of reward presentation. These results also indicate that caution is needed when extrapolating findings of the impact of perinatal and/or short-term HF feeding on the persistence of offspring behaviour beyond early adulthood, and highlights the need for more studies into the long-term effects of perinatal diet across the lifespan of the offspring.

## Figures and Tables

**Figure 1 nutrients-14-05161-f001:**
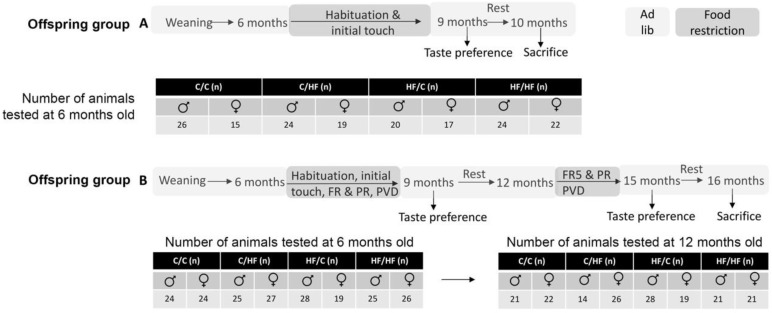
Schematic of offspring testing protocol and schedule. Offspring in group A underwent a shortened testing protocol and tissues were collected at 10 months of age. Animals in group B underwent a longitudinal testing protocol and tissues were collected at 16 months of age.

**Figure 2 nutrients-14-05161-f002:**
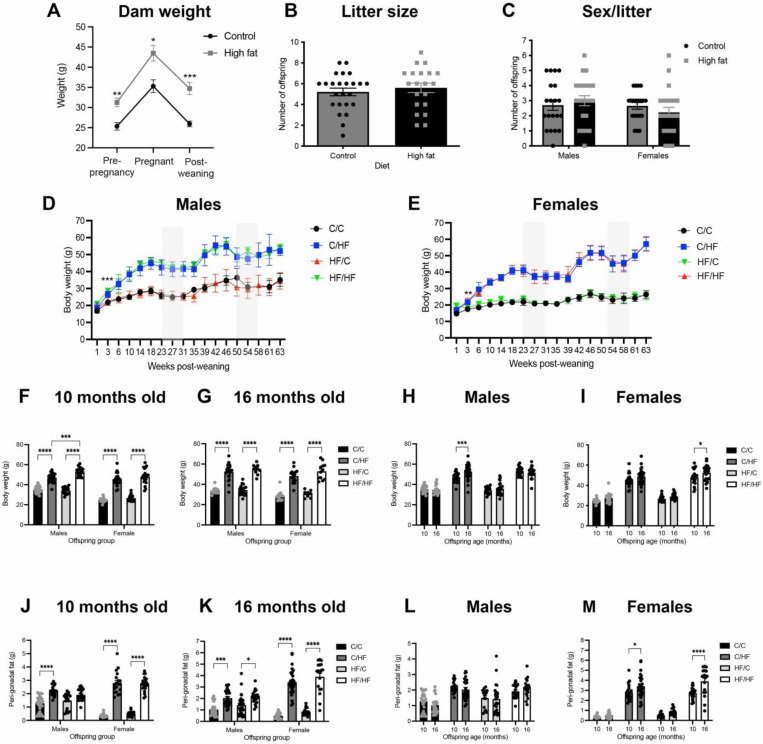
Phenotypes of animals fed the C and HF diets. (**A**) Body weight of dams fed control (*n* = 12) or high fat diet (*n* = 11) after 4 weeks of feeding (pre-pregnancy), during gestation and post-weaning of offspring. * *p* < 0.05, ** *p* < 0.01, *** *p* < 0.001, two-way ANOVA with Sidak’s post hoc. (**B**,**C**) Total number of offspring per litter (**B**) and sex balance of male and female offspring (**C**) between dams fed the control (**C**, *n* = 24) and high fat (**H**,**F**, *n* = 20) diets. (**D**,**E**) Body weight gain after weaning over the life course of a subset of male (**D**) and female (**E**) offspring (*n* = 9–12/group). Note that error bars represent standard deviations so that they can be visualized. Shaded regions represent periods of food restriction during behavioural testing. Here, ** *p* < 0.01, *** *p* < 0.001, mixed-effects ANOVA with Sidak’s post-hoc. (**F**–**I**) Body weight of male and female C/C, C/HF, HF/C, and HF/HF offspring at 10- (**F**) and 16-months of age (**G**), and between 10- and 16-months of age in male (**H**) and female offspring (**I**). Here, * *p* < 0.05, *** *p* < 0.001, **** *p* < 0.0001, one-way ANOVA with Sidak’s post hoc. (**J**–**M**) Weight of peri-gonadal fat from 10-(**J**) and 16-month-old (**K**) offspring and age-related change in fat weight of male (**L**) and female (**M**) offspring. Here, ** *p* < 0.01, *** *p* < 0.001, **** *p* < 0.0001, one- or two-way ANOVA with Sidak’s post hoc. For body weight measures of 10-month-old offspring, *n* = 19–26 males and *n* = 16–23 females; for 16-month-old offspring, *n* = 11–22 males and *n* = 8–19 females.

**Figure 3 nutrients-14-05161-f003:**
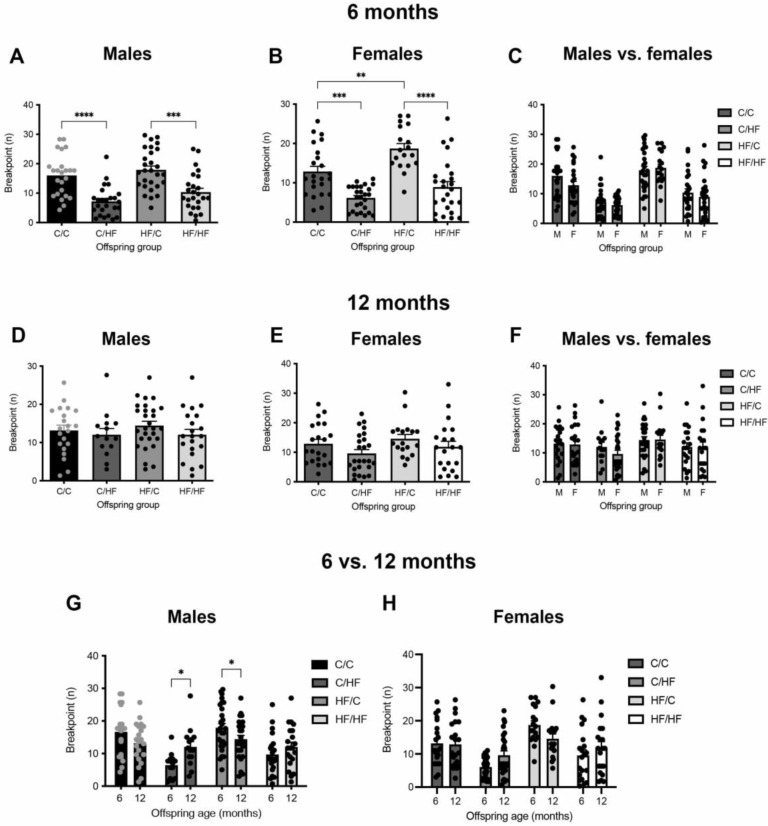
Effect of peri- and postnatal C and HF diet on motivation in young and aged offspring. (**A–C**) Breakpoint of 6-month-old male (**A**) and female (**B**) offspring, and comparison between male and female breakpoints (**C**) on the PR2 task. Males *n* = 24–28, females *n* = 19–27, ** *p* < 0.01, *** *p* < 0.001, **** *p* < 0.0001, one- or two-way ANOVA with Sidak’s post hoc. (**D**–**F**) Breakpoint of the same male (**D**) and female I(**E**) offspring groups re-tested at 12 months of age and comparison of breakpoints between 12-month-old male and female offspring (**F**). Males *n* = 14–28, females *n* = 19–26, one- or two-way ANOVA with Sidak’s post hoc. (**G**,**H**) Comparison of breakpoint of between 6- and 12 months of age in male (**G**) and female (**H**) offspring. Here, * *p* < 0.05, two-way, repeated measures ANOVA with Sidak’s post hoc.

**Figure 4 nutrients-14-05161-f004:**
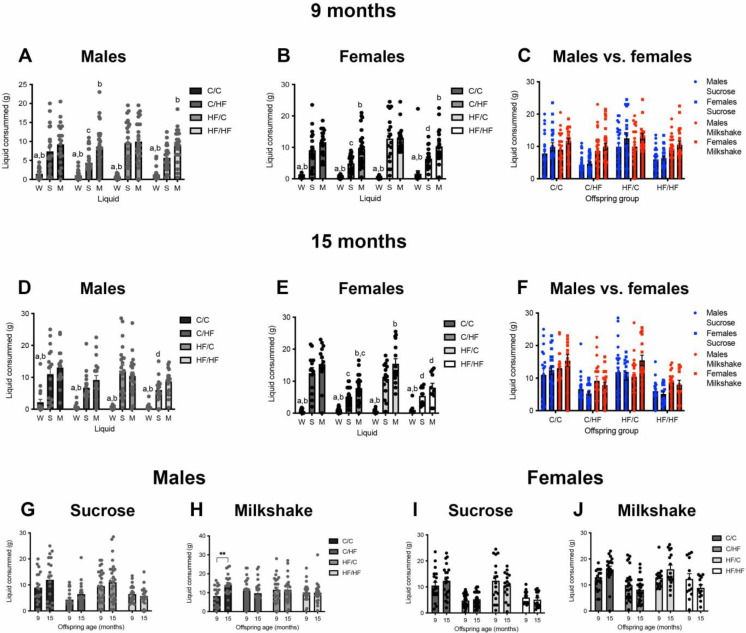
Taste preference of young and aged offspring. (**A**–**C**), 24 h intake of water, sucrose, and milkshake by 9-month-old male (**A**) and female (**B**) offspring, and comparison of intake between males vs. females (**C**). Males *n* = 22–30, females *n* = 18–27. Here, a = *p* < 0.05 vs. milkshake, b = *p* < 0.05 vs. sucrose, c = *p* < 0.05 vs. C/C, and d = *p* < 0.05 vs. HF/C, with two-way, repeated measures ANOVA with Sidak’s post hoc. (**D**–**F**), Intake of water, sucrose, and milkshake by the same male (**D**) and femaIe (**E**) animals at 15 months of age and comparison between male vs. female offspring (**F**). *n* = 17–22 males, *n* = 14–27 females. Here, a = *p* < 0.05 vs. milkshake, b = *p* < 0.05 vs. sucrose, c = *p* < 0.05 vs. C/C, and d = *p* < 0.05 vs. HF/C, two-way, with repeated measures ANOVA with Sidak’s post hoc. (**G**–**K**), Comparison of sucrose (**G**,**I**) and milkshake (**H**,**J**) consumption between 9- and 15-month-old male (**G**,**H**) and female (**I**,**J**) offspring. Here, ** *p* < 0.01, two-way, repeated measures ANOVA with Sidak’s post hoc. Abbreviations are as follows: W = water, S = sucrose, and M = milkshake.

**Figure 5 nutrients-14-05161-f005:**
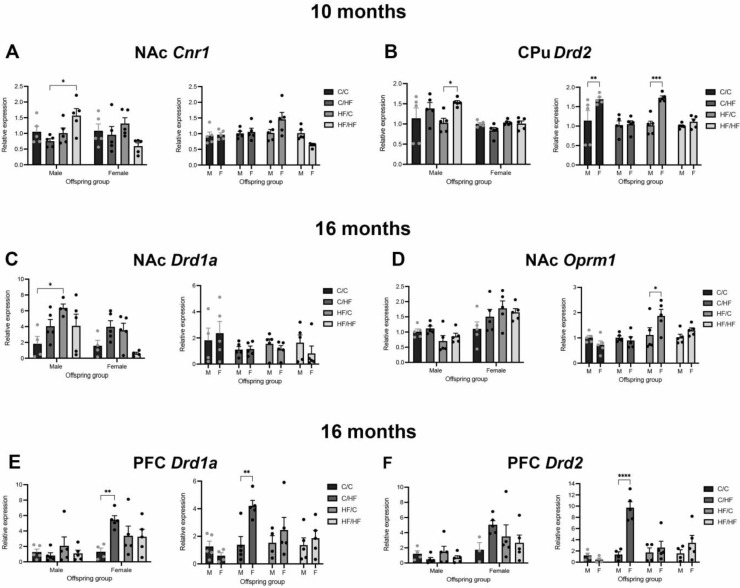
Relative expression of markers of dopamine, μ-opioid, and endocannabinoids in the brains of 10-month-old offspring. (**A**,**B**) Relative mRNA levels of *Crn1* in the nucleus accumbens (NAc, **A**) and *Drd2* in the caudate–putamen (CPu, **B**) of 10-month-old male and female offspring groups. (**C**–**F**) mRNA levels of *Drd1a* (**C**,**E**), *Oprm1* (**D**), and *Drd2* (**F**) in the nucleus accumbens (**C**,**D**), and prefrontal cortex (PFC, **E**,**F**) of 16-month-old male and female offspring. Here, *n* = 4–5 for all groups. For comparisons between diet groups, mRNA levels are expressed relative to the male (M) or female (F) C/C group. For sex comparisons, levels are expressed relative to the males in each diet group. Here, * *p* < 0.05, ** *p* < 0.01, *** *p* < 0.001, and **** *p* < 0.0001, with one- or two-way ANOVA with Sidak’s post hoc.

**Figure 6 nutrients-14-05161-f006:**
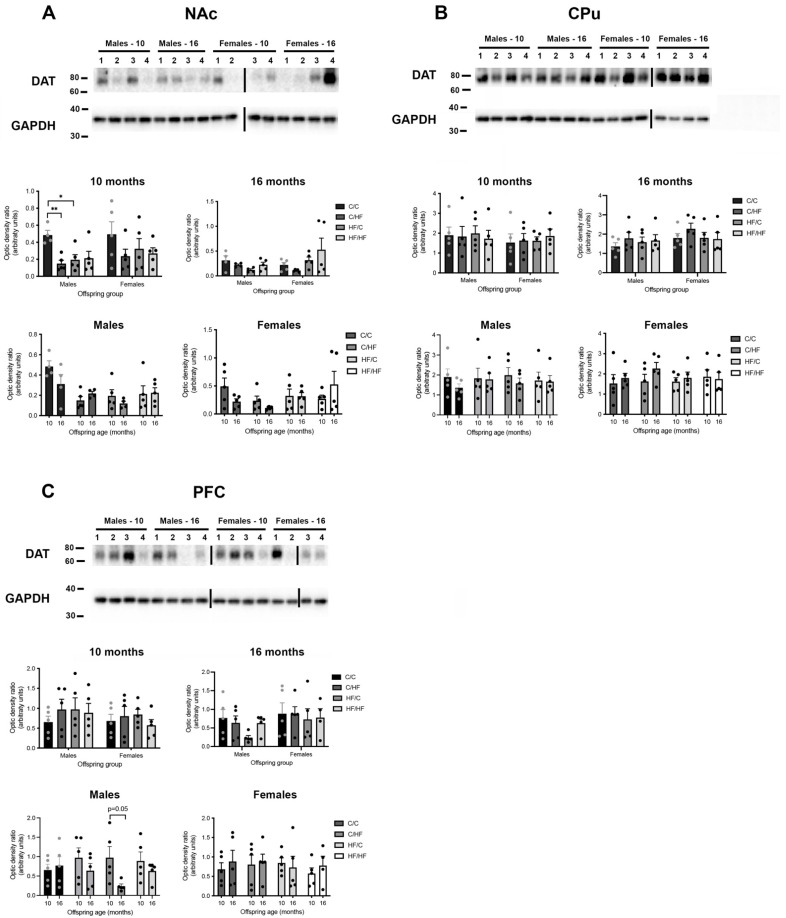
Evaluation of protein levels of dopamine transporter in young and aged offspring. Western blots and quantification of protein levels of dopamine transporter (DAT) in the nucleus accumbens (NAc, **A**), caudate–putamen (CPu, **B**), and prefrontal cortex (PFC, **C**) of 10- and 16-month-old male and female offspring groups. Black lines show loading of samples on different gels. Here, 1 = C/C, 2 = C/HF, 3 = HF/C, and 4 = HF/HF. Numbers following ‘males’ and ‘females’ represent age of animals in months. Here, *n* = 5 for all groups; * *p* < 0.05 and ** *p* < 0.01, for one-way or two-way ANOVA with Sidak’s post hoc.

**Figure 7 nutrients-14-05161-f007:**
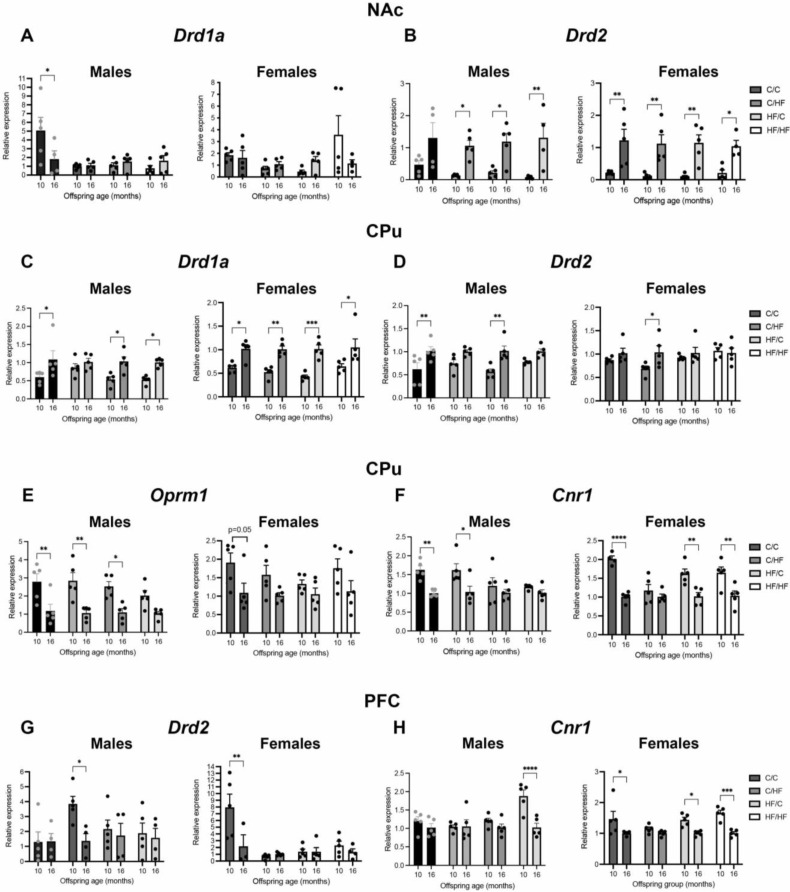
Expression of markers of dopamine, μ-opioid, and endocannabinoids in the brains of 10- vs. 16-month-old offspring. (**A**,**B**) Relative mRNA levels of *Drd1a* (**A**,**C**), *Drd2* (**B**,**D**,**G**), *Oprm1* (**E**), and *Cnr1* (**F**,**H**) in the nucleus accumbens (NAc, **A**,**B**), caudate–putamen (CPu, **C**–**F**), and prefrontal cortex (PFC, **G**,**H**) of 10- and 16-month-old male and female offspring groups. The mRNA levels are expressed relative to levels in 16-month-old animals in each diet group. Here, *n* = 4–5 for all groups; * *p* < 0.05, ** *p* < 0.01, *** *p* < 0.001, **** *p* < 0.0001, for two-way ANOVA with Sidak’s post hoc.

**Table 1 nutrients-14-05161-t001:** Summary of body weight, motivation, and taste preference in young (6–10 months old) and aged (12–16 months old) male and female offspring. Here, ↑ = higher in specified diet group; ↓ = lower in specified diet group; - = no difference between diet groups. Abbrevations are as follows: S = sucrose; M = milkshake.

	Males					Females			
6–10 Months Old		C/C vs. C/HF	C/C vs. HF/C	C/HF vs. HF/HF	HF/C vs. HF/HF	C/C vs. C/HF	C/C vs. HF/C	C/HF vs. HF/HF	HF/C vs. HF/HF
	Body weight	↓ C/C	-	↓ C/HF	↓ HF/C	↓ C/C	-	-	↓ HF/C
	Motivation	↑ C/C	-	-	↑ HF/C	↑ C/C	↓ C/C	-	↑ HF/C
	Sucrose preference	S: ↑ C/C M: -	S: - M: -	S: - M: -	S: - M: -	S: ↑ C/C M: -	S: - M: -	S: - M: -	S: ↑ HF/C M: -
12–16 months old	Body weight	↓ C/C	-	-	↓ HF/C	↓ C/C	-	-	↓ HF/C
	Motivation	-	-	-	-	-	-	-	-
	Sucrose preference	S: - M: -	S: - M: -	S: - M: -	S: ↑ HF/C M: -	S: ↑ C/C M: ↑ C/C	S: - M: -	S: - M: -	S: ↑ HF/C M: ↑ HF/C

**Table 2 nutrients-14-05161-t002:** Summary of main changes in gene or protein expression in the NAc, CPu, and PFC of 16- vs. 10-month-old male and female offspring. Here, ↑ = increased; ↓ decreased; - = no change.

		C/C		C/HF		HF/C		HF/HF	
Brain Region	Gene or Protein	Males	Females	Males	Females	Males	Females	Males	Females
NAc	*Drd1a*	↓	-	-	-	-	-	-	-
	*Drd2*	↑	↑	↑	↑	↑	↑	↑	↑
	*Crn1*	-	-	-	-	-	-	-	-
	*Oprm1*	-	-	-	-	-	-	-	-
	DAT	-	-	-	-	-	-	-	-
CPu	*Drd1a*	↑	↑	-	↑	↑	↑	↑	↑
	*Drd2*	↑	-	-	↑	↑	-	-	-
	*Crn1*	↓	↓	↓	-	-	↓	-	↓
	*Oprm1*	↓	↓	↓	-	↓	-	-	-
	DAT	-	-	-	-	-	-	-	-
PFC	*Drd1a*	-	-	-	-	-	-	-	-
	*Drd2*	-	↓	↓	-	-	-	-	-
	*Crn1*	-	↓	-	-	-	↓	↓	↓
	*Oprm1*	-	-	-	-	-	-	-	-
	DAT	-	-	-	-	↓	-	-	-

## Data Availability

The datasets used and/or analysed during the current study are available from the corresponding author on reasonable request.

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
