# Peer review of "Appetitive Motivation and Associated Neurobiology Change Differentially across the Life Course of Mouse Offspring Exposed to Peri- and Postnatal High Fat Feeding"

_nutrients, 2022, doi:10.3390/nu14235161_

Round 1
Reviewer 1 Report
In this manuscript, the authors evaluated and demonstrated the longitudinal effect of perinatal and chronic postnatal high fat intake on appetitive motivation in young and aged offspring. The study design is scientifically sound, and manuscript is well written in general. Minor revisions should be focused on making it clearer of the rationale of measuring the gene expressions in the Introduction and the take home message from the whole study. Please find my other suggestions for improvements as follows.
1. What is the age of the female breeder mice used in the study?
2. It is better to spell out the full name of the gene being analyzed in the Abstract.
3. Line 68-71, the main point of this summary is not very clear to me. Please consider reorganizing since this is part of the rationale of measuring the gene expressions.
4. Consider making figure 1A a standalone figure so that it can be moved closer to the Method section to assist readers to understand the study design.
5. PCR – the amount of total RNA used, and the condition and primer sequences used in the PCR reactions all need to be specified.
6. Table 1 is a good summary of the changes. Formatting needs to be improved to make it clearer.
Author Response
- Minor revisions should be focused on making it clearer of the rationale of measuring the gene expressions in the Introduction and the take home message from the whole study.
The Introduction has been expanded to include more information about the role of dopamine, opioid and endocannabinoids in the regulation of appetitive motivation. The main conclusions from the study have been added to the abstract.
- What is the age of the female breeder mice used in the study?
Dams were 18- 27 weeks old at mating. This information has been added to the Methods section (line 152).
- It is better to spell out the full name of the gene being analyzed in the Abstract.
This information has now been added to the Abstract.
- Line 68-71, the main point of this summary is not very clear to me. Please consider reorganizing since this is part of the rationale of measuring the gene expressions.
As indicated above, the Introduction has been expanded to include more information about the role of dopamine, opioid and endocannabinoids in the regulation of appetitive motivation.
- Consider making figure 1A a standalone figure so that it can be moved closer to the Method section to assist readers to understand the study design.
The figure of the experimental design has now been made into standalone figure (Fig. 1).
- PCR – the amount of total RNA used, and the condition and primer sequences used in the PCR reactions all need to be specified.
Additional information about the RT-qPCR has now been added to the Methods (lines 235 to241 and Supplemental Table 1).
- Table 1 is a good summary of the changes. Formatting needs to be improved to make it clearer.
Additional columns have been added to the gene summary table (now Table 2) to improve clarity of the interpretation.
Reviewer 2 Report
This study documented the longitudinal effect of perinatal and chronic postnatal HF intake on appetitive motivation in young and aged offspring. Female C57Bl/6 were fed either a control (C) or high fat (HF) diet before mating through to lactation. At weanng, offspring were maintained on the C or HF diet, generating 4 diet groups: C/C, C/HF, HF/C and 16 HF/HF based on the pre/post weaning diet. Motivation for an appetitive reward and taste preference as well essential genes for dopaminergic pathways were measured between 6-16 months mice in both sexes. These results have significant implications for patterns of hedonic eating across the life course and the relative risk of obesity at different timepoints.
Previous reports are not consistent or even controversial for several reasons, and more focused on younger offspring. This manuscript is helpful to fill the gap and will interest in researchers studying the effect of maternal obesity on the health of offspring.
This manuscript is very descriptive which contains large amount of data. The major challenge is the organization of the data to clearly present the differences among 4 groups, age association and sex, in addition, gene profiles in three different brain regions. Another challenge is the mechanism association. the current discussion is too general that we can not get a clear conclusion, like which gene in which brain region is linked to which phenotype in which group/sex at which age. The writing can be improved in these two aspects.
Other minor suggestions:
Figure 3B, letters are buried in the dots.
Bar graphs, some dots are invisible in dark bars. change color or style (eg, empty circle instead of solid dots)
A summary table for metabolic phenotypes, motivation and taste preference.
Additional information for the dams and pups before weaning: Miscarriage rate or successful delivery rate. Pup survival rate.
More details for methods. E.g., it looks that mice are group housed before sucrose tests. How many litters were they? Were they grouped by litters or random after weaning? Have you randomized the body weight and dams at weaning for grouping (C/C, C/HF, HF/C and 16 HF/HF)? E.g., if a group of offspring are all from low litter number dams, this will produce bias before all the tests. How many mice per cage? Did you measure group food intake? How many days of single housing before sucrose test? Single house itself is a stressor that needs almost 1 week to recover.
Author Response
- This manuscript is very descriptive which contains large amount of data. The major challenge is the organization of the data to clearly present the differences among 4 groups, age association and sex, in addition, gene profiles in three different brain regions. Another challenge is the mechanism association. the current discussion is too general that we can not get a clear conclusion, like which gene in which brain region is linked to which phenotype in which group/sex at which age. The writing can be improved in these two aspects.
More subheadings have been added to the Results section to help clarify the presentation of the data. As outlined in the Introduction, appetitive motivation results from coordinated communication between DA, opioid and eCB signals within the mesolimbic and mesocortical pathways, rather than localized changes in individual genes. In the Discussion, we have highlighted the following observations and outstanding questions related to the impact of HF feeding on motivation and associated neurobiology:
- Lines 1050-1057: Higher levels of Drd2 mRNA were found in the CPu of 10-month-old HF/C females vs. HF/C males in association with higher motivation in HF/C females. We hypothesized that increased motivation may be due to increased DA release following burst firing of dopaminergic neurons in response to D2R desensitization in the CPu. More experimental work is needed to confirm this hypothesis.
- Lines 1081-1090: Decreased DAT expression was found in the NAc of 10-month-old C/HF and HF/HF mice that showed decreased appetitive motivation, suggesting that the lower motivation may be due to decreased DA concentration in these animals.
- Lines 1111-1116: The observed age-related shift towards decreased motivation in the 12-month-old C/C mice is consistent with an age-related decline in reduced striatal D2R availability.
- Lines 1131-1143: The age-related increase in Drd1a and Drd2 gene expression in the CPu in C/HF and HF/HF animals may have induced a sensitization to the milkshake reward that contributed to the increased motivation in these animals at 16-month of age. Additional work is needed to determine why a similar increase in Drd2expression in the NAc and increased Drd1a in the CPu were observed across all diet groups in which motivational changes are opposite.
Other minor suggestions:
- Figure 3B, letters are buried in the dots.
The formatting of the figure (Fig. 4) has been corrected.
- Bar graphs, some dots are invisible in dark bars. change color or style (eg, empty circle instead of solid dots)
Graphs have been reformatted throughout to enhance the contrast of the individual data points.
- A summary table for metabolic phenotypes, motivation and taste preference.
Table 1 summarising the body weight, motivation and taste preference results has now been added in the Results section.
- Additional information for the dams and pups before weaning: Miscarriage rate or successful delivery rate. Pup survival rate.
Successful delivery rate was 88% for C-fed dams and 80% for HF-fed dams. 2 litters from HF-fed dams were cannibalized but all other pups survived. This information has been added to Methods (lines 155-156).
- More details for methods. E.g., it looks that mice are group housed before sucrose tests. How many litters were they? Were they grouped by litters or random after weaning? How many mice per cage? Have you randomized the body weight and dams at weaning for grouping (C/C, C/HF, HF/C and 16 HF/HF)? E.g., if a group of offspring are all from low litter number dams, this will produce bias before all the tests.
Offspring were group housed by litter with an average of n=4 per cage (range of n=2-6 mice/cage) throughout the experiment, with the exception of 4 days of single housing during sucrose preference testing. Offspring from n=34 litters from C-fed dams and from n=39 litters of HF-fed dams were used in the experiments. Offspring from these litters were randomly allocated to C or HF diet. This information has been added to the Methods section (lines 158-162).
- Did you measure group food intake?
Food intake and calorie consumption has been previously published (Contu et al. Front Neurosci. 2019 13:1045) as indicated in the text (line 297).
- How many days of single housing before sucrose test? Single house itself is a stressor that needs almost 1 week to recover.
Mice were singly housed for 24 hours before starting the sucrose preference test, which took place over 4 days, including normal water intake. This information has been added to the Methods (line 221). Due to ethical restrictions, we are not able to singly house mice for more than 5 days.
Reviewer 3 Report
The manuscript prepared by Contu et al. examined the effects of a high-fat diet during perinatal and postweaning periods on appetitive motivation and neurochemical changes at different time points. The subject is important and the experiments were well conducted. There are several minor issues that should be addressed.
1. For the offspring, were they fed with the control or high-fat diet after weaning until sacrifice? It is better to show the curves of body weight during the life course.
2. The time for behavioral tests and tissue collection should be provided. The circadian rhythm may affect the appetite and neurochemistry.
3. The emphasis of the report should be more clear. The authors showed the differences between time points and between genders. They may discuss these items separately.
The rationales for analyzing dopamine, opioid, and CB1 receptors should be better elaborated. How do the expression of these (and other) receptors in the examined (and other) brain areas account for the appetitive behaviors? A hypothetical cell-type-based neural circuit may be proposed.
Author Response
- For the offspring, were they fed with the control or high-fat diet after weaning until sacrifice? It is better to show the curves of body weight during the life course.
As indicated in the text, offspring were fed the C or HF diet from weaning until sacrifice (line 161). Graphs showing body weights after weaning over the life course of a subset of offspring has now been added (Fig. 2D and E).
- The time for behavioral tests and tissue collection should be provided. The circadian rhythm may affect the appetite and neurochemistry.
Behavioral testing took place during the light phase, between 10:00 and 15:00. Tissue collection was also conducted during the light phase. Although exact timings were not recorded, this would typically have been between 09:30 and 15:30. This information has been added to the Methods (lines 209 and 228).
- The emphasis of the report should be more clear. The authors showed the differences between time points and between genders. They may discuss these items separately.
More subheadings have been added to the Results section to help organize the presentation of the data. A paragraph describing the difference between males and females has been added to the Discussion.
- The rationales for analyzing dopamine, opioid, and CB1 receptors should be better elaborated. How do the expression of these (and other) receptors in the examined (and other) brain areas account for the appetitive behaviors? A hypothetical cell-type-based neural circuit may be proposed.
The Introduction has been expanded to include more information about the role of dopamine, opioid and endocannabinoids and the underlying anatomical pathways in the regulation of appetitive motivation.